# STRUCTURED GRAPH REDUCTION FOR EFFICIENT GNN

## ABSTRACT

Scalability remains a prominent challenge for Graph Neural Networks (GNNs) when dealing with large-scale graph data. Graph coarsening is a technique that reduces a large graph to a smaller tractable graph. A good quality graph representation with specific properties is needed to achieve good performance with downstream applications. However, existing coarsening methods could not coarsen graphs with desirable properties, such as sparsity, scale-free characteristics, bipartite structure, or multi-component structure. This work introduces a unified optimization framework for learning coarsened graphs with desirable structures and properties. The proposed frameworks are solved efficiently by leveraging block majorization-minimization, $\log$ determinant, structured regularization, and spectral regularization frameworks. Extensive experiments with real benchmark datasets elucidate the proposed framework's efficacy in preserving the structure in coarsened graphs. Empirically, when there is no prior knowledge available regarding the graph's structure, constructing a multicomponent coarsened graph consistently demonstrates superior performance compared to state-of-the-art methods.

## 1 INTRODUCTION

Graph machine learning is a common tool for modeling and analyzing complex systems, such as social networks, biological networks, transportation networks, and computer networks,etc.(Battaglia et al., 2018; Wu et al., 2020; Zhou et al., 2020; Bruna et al., 2013; Chen et al., 2020b; Defferrard et al., 2016). Large graphs are becoming increasingly common, requiring significant computational resources for data loading and processing. As a result, analyzing and scaling up graph-based machine learning becomes challenging due to the bottleneck imposed by these large graphs(Rong et al., 2019; Chen et al., 2020a). Techniques such as graph reduction or coarsening (Loukas, 2019; Kumar et al., 2022; Chen et al., 2022), summarization (Liu et al., 2018; Riondato et al., 2017) or condensation (Jin et al., 2021; 2022), graph sparsification (Fung et al., 2011; Spielman & Teng, 2011), etc. have emerged as promising approaches to address this issue. These techniques aim to coarsen or reduce given graphs into smaller ones, allowing for more efficient analysis and processing of the data.

For building effective graph-based approaches, the choice of graphs to be used for encoding the relationships is a critical decision and often more important than the particular algorithm or type of loss function to be used. This becomes more critical when the downstream tasks are performed over the reduced graph. For better performance, graphs with additional properties (e.g., structures) are needed for interpretability and precise identification of the relationships in these data sets. There are plenty of works on learning structured graphs from data, for example, for bipartite graph learning (Narang & Ortega, 2012), scale-free (Liu & Ihler, 2011), sparse(Yuan & Lin, 2007), and multicomponent (Hao et al., 2018) graphs. These methods are computationally heavy and can learn only a specific type of structure. Recently, the work in (Kumar et al., 2020) has developed an optimization-based framework, where with a suitable choice of regularization it can learn graphs with a variety of important structures.

However, how to enforce desirable structure while learning a reduced graph is not well understood yet. There are two distinct approaches to enforcing structure in a coarsened graph. The first approach involves a two-step process: initially coarsening the graph and subsequently applying existing algorithms designed to enforce structural constraints. In contrast, the second approach simultaneously integrates the coarsening and structural enforcement steps. The joint learning approach works better

because it leverages the synergy between coarsening and structural enforcement, leading to a more adaptive, informed, and data-driven optimization process that ultimately results in a coarsened graph with improved structural properties and, consequently, better overall performance. In this work, we have introduced a novel optimization-based framework for learning coarsened graphs with desirable structure and properties. In this work, we have enforced four structures: sparse, scale-free, multi-component, and bipartite in the coarsened graph. The formulated problems for obtaining sparse and scale-free coarsened graphs are convex optimization problems, while for obtaining multi-component and bipartite coarsened graphs are multi-block non-convex optimization problems which are solved efficiently by leveraging block majorization-minimization, $\log$ determinant, Laplacian and adjacency spectral constraints, and regularization frameworks. The developed algorithm is convergent and enforces the desired properties in the learned coarsened graph.

We have applied the proposed coarsening algorithms to real datasets for node classification tasks and compared them to the recent graph coarsening techniques. By enforcing structure in the coarsened graph, we observed a notable increase in accuracy compared to the state-of-the-art methods. Furthermore, we have also utilized proposed algorithms to perform classification on various GNN architecture like GCN (Kipf & Welling, 2016), APPNP (Gasteiger et al., 2018), and GAT (Velivcković et al., 2017), respectively. Also, the proposed structured graph coarsening methods are faster than the state-of-the-art graph coarsening methods. Extensive experiments elucidate the efficacy of the proposed framework for real-world applications.

In this work, we have investigated different structure plays a different role in performing the downstream task using graph neural network. Prior knowledge about a graph's structure can benefit coarsening based on that specific structure. In the absence of prior knowledge about a graph's structure, the multiple component method typically proves to be a robust approach, and empirical evidence consistently demonstrates the superior performance of the Multicomponent Coarsened Graph Learning (MGC) algorithm compared to state-of-the-art alternatives. Within the framework of multiple component coarsened graphs, the key strategy involves partitioning the graph into components, aligning their number with the classes present in the original graph. This approach effectively simplifies graph analysis and often reveals meaningful insights, making it a valuable choice.

## 2 BACKGROUND AND PROPOSED FORMULATION

In this section, we review the graph coarsening method and formulate the problem of structured graph coarsening.

### 2.1 GRAPH COARSENING

Given an original graph $\mathcal{G} = (V, E, X \in \mathbb{R}^{p \times d}, Y \in \mathbb{R}^{p \times l})$ with $p$ nodes, the goal of graph coarsening is to construct an appropriate "smaller" or coarsen graph $\mathcal{G}_c = (\tilde{V}, \tilde{E}, \tilde{X} \in \mathbb{R}^{k \times d}, \tilde{Y} \in \mathbb{R}^{k \times l})$ with $k << p$ nodes. Given the Laplacian matrix $\Theta \in \mathbb{R}^{p \times p}$ and adjacency matrix $A \in \mathbb{R}^{p \times p}$ corresponding to the large graph, the coarsened graph Laplacian and adjacency matrices are $\Theta_c \in \mathbb{R}^{k \times k}$ and $A_c \in \mathbb{R}^{k \times k}$ are obtained using the relation $\Theta_c = C^\top \Theta C$ and $A_c = C^\top A C$ (Loukas, 2019) where $C \in \mathbb{R}^{p \times k}$ is the mapping matrix that maps nodes of original graph to supernode of the coarsened graph. For a valid coarsening, the mapping matrix should belong to the following set (Kumar et al., 2023)

$$\mathcal{C} = \left\{ C \geq 0 | \langle C_i, C_j \rangle = 0 \,\forall\, i \neq j, \quad \langle C_i, C_i \rangle = d_i, \|C_i\|_0 \geq 1 \text{ and } \left\| [C^\top]_i \right\|_0 = 1 \right\} \quad (1)$$

where $C_i$, $C_j$ and $[C^\top]_i$ represents the $i-$th column, $j-$th column and $i$-th row of mapping matrix $C$ respectively. Many graph coarsening algorithms learn the mapping matrix $C$, and using $C$, we can obtain the Laplacian matrix of a coarsened graph $\Theta_c = C^\top \Theta C$.

### 2.2 GRAPH COARSENING FOR SCALING UP GNN

The coarsened graph serves as the foundation for downstream tasks, such as node classification, where a Graph Neural Network (GNN) is trained using this coarsened graph $\mathcal{G}_c = (\tilde{V}, \tilde{E}, \tilde{X} = C^\dagger X, \tilde{Y} = argmax(C^\dagger Y))$ (Huang et al., 2021). The effective training of Graph Neural Networks

(GNNs) relies significantly on the labels $\tilde{Y}$ assigned to the coarsened graph. In this context, each supernode's label is determined by selecting the most frequent class among its constituent nodes that share the same class. For a given original graph $\mathcal{G}$, there are various possibilities of the coarsened graph $\mathcal{G}_c$ and the loading matrix $C$. To quantify the quality of a coarsened graph in terms of nodes of the original graph having the same label mapped to the supernode of the coarsened graph relies on the $\phi$ matrix that is defined as:

**Definition 2.1.** A loading matrix $C$ is considered balanced, and a coarsened graph is considered informative when, after transforming the one-hot matrix $Y \in \mathbb{R}^{p \times l}$ of labels from the original graph $\mathcal{G}$ using $C$, the resulting matrix $\phi = C^\top Y$ exhibits sparsity in its rows (Ghoroghchian et al., 2021).

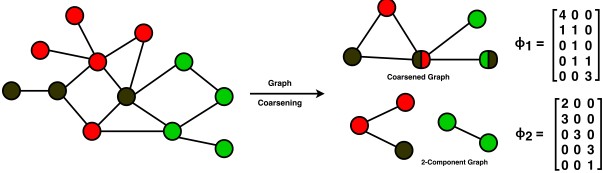

Figure 1: This figure illustrates that multiple structures can be imposed on the resulting coarsened graph for a given original graph $\mathcal{G}$. However, the optimal training for Graph Neural Networks (GNNs) using a coarsened graph is attained when the $\phi$ matrix exhibits only one non-zero entry in each row. This essentially means that the rows of the $\phi$ matrix should ideally be as sparse as possible. In the provided example, it's evident that the rows of $\phi_2$ matrix exhibit a greater degree of sparsity compared to rows of $\phi_1$. Consequently, the coarsened graph associated with the $\phi_2$ matrix is better suited for training GNNs.

Moreover, the recent graph coarsening methods for example, (Loukas & Vandergheynst, 2018) is a heuristic-based approach, (Kumar et al., 2023) Dirichlet energy optimization-based approach, and (Jin et al., 2020) is a deep learning method for learning the mapping matrix $C$. Recent state-of-the-art techniques often encounter challenges when attempting to learn a sparse $\phi$ matrix, making them less suitable for downstream tasks like node classification. The task of learning a coarsened graph with a sparse $\phi$ matrix is known to be computationally demanding and falls into the realm of combinatorial hard problems.

To address this challenge effectively, our approach involves a two-step process. First, we enforce specific structural characteristics such as multi-component, bipartite, sparsity, or scale-free properties within the coarsened graph. Subsequently, we calculate the $\phi$ matrix for each of these cases. We train our Graph Neural Network (GNN) using the coarsened graph associated with a sparser $\phi$ matrix. Subsequently, during the testing phase, we conduct evaluations on the original graph. Importantly, our empirical findings consistently indicate that the coarsened graph with a sparser $\phi$ matrix outperforms other configurations. For instance, as illustrated in Figure 1, the coarsened graph corresponding to the $\phi_2$ matrix exhibits superior performance when training the Graph Neural Network (GNN). The figure below illustrates the workflow of our work.

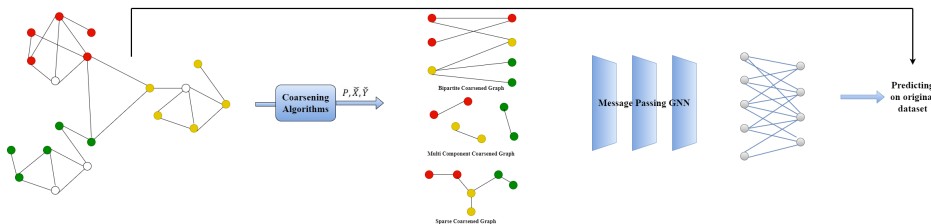

Figure 2: The figure shows an illustration of node classification of an original graph $\mathcal{G}$ using coarsened graph $\mathcal{G}_c$.

## 3    PROPOSED FRAMEWORK FOR STRUCTURED GRAPH COARSENING

The proposed optimization-based framework for learning a structured coarsened graph is:

$$\begin{aligned}
&\underset{C, \Theta_c}{\text{minimize}} && -\gamma \log \text{gdet}(\Theta_c) + \frac{\lambda}{2}\|C^\top\|_{1,2}^2 + \alpha h(\Theta_c) \\
&\text{subject to} && C \in \mathcal{S}_c = \left\{ C \geq 0, \ \|[C^\top]_i\|_2^2 \leq 1 \, \forall \, i = 1, 2, 3, \ldots, p \right\}, \lambda(\mathcal{T}(\Theta_c)) \in \mathcal{S}_\lambda
\end{aligned} \tag{2}$$

where $\mathrm{gdet}(\Theta_c)$ denotes the generalized determinant defined as the product of the non-zero eigenvalues of the coarsened graph Laplacian matrix $\Theta_c$, $h(\Theta_c)$ is the regularizer, $\lambda(\Theta_c)$ denotes the eigenvalues of $\Theta_c$, $\mathcal{S}_\lambda$ is the set containing spectral constraint on the eigenvalues, $\mathcal{T}(\cdot)$ is a linear operator used to enforce eigenvalue constrained on other than coarsened Laplacian matrix, and $\gamma, \lambda$, and $\alpha > 0$ are the hyperparameters. Moreover, $\mathcal{S}$ enforces the structure on the coarsened graph to be learned (Kumar et al., 2020). Next, we will introduce different choices of $\mathcal{S}_\lambda$ and $h(\Theta_c)$ that will enforce different structures in the resulting coarsened graph.

- **Sparse coarsened graph** can be learned using the following regularizer

$$h(\Theta_c) = \|C^\top \Theta C\|_F^2 \tag{3}$$

- **Multi-component coarsened graph** where the super-node set can be partitioned into $n$ disjoint subsets are having the first $n$ eigenvalues of the Laplacian matrix as zeros. Thus the eignevalue constrained of it's Laplacian are expressed as (Kumar et al., 2020):

$$\mathcal{S}_\lambda = \left\{ \{\lambda_j = 0\}_{j=1}^n, c_1 \leq \lambda_{n+1} \leq \ldots \leq \lambda_k \leq c_2 \right\} \tag{4}$$

Where $n \geq 1$ denotes the number of connected components in the learned coarsened graph, and $c_1, c_2 > 0$ are constants that depend on the number of edges and their weights.

- **Bipartite Coarsened graph**: A coarsened graph is bipartite if and only if the eigenvalues of the adjacency matrix of the coarsened graph $A_c = C^\top A C$ is symmetric about the origin (Kumar et al., 2020) such that:

$$\mathcal{S}_\psi = \{\psi_1 \geq \psi_2 \geq \ldots \psi_{k-1} \geq \psi_k, \psi_i = -\psi_{k-i+1}, i = 1, 2, \ldots k\} \tag{5}$$

- **Scale Free Coarsened Graph**: Scale-free graphs represent a class of graphs that follow a power law in their degree distribution (Liu & Ihler, 2011), i.e, the degree of a node $i$ follows a power law degree distribution $p(d) \propto d^{-\alpha}$, where $\alpha > 0$. To enforce the scale-free structure in the resulting coarsen graph, we will use the regularizer $h(\Theta_C)$ at $t$-th iteration (Liu & Ihler, 2011):

$$h(\Theta_c) = \sum_{i \neq j} \delta_{ij}^t |[C^\top \Theta C]_{ij}| + \beta \sum_i |[C^\top \Theta C]_{ii}| = \|C^\top A C \delta \mathbf{1}_{k \times 1}\|_1 + \|C^\top D C \beta \mathbf{1}_{k \times 1}\|_1 \tag{6}$$

where $\delta_{ij}^t = \alpha \left( \dfrac{1}{\sum_{i \neq j} |\Theta_{cij}^t| + \epsilon_i} + \dfrac{1}{\sum_{j \neq i} |\Theta_{cij}^t| + \epsilon_j} \right)$ \hfill (7)

where $\beta = 2\frac{\alpha}{\epsilon_i}$, $\Theta_c^t = [[C^\top \Theta C]^t]$ is the estimate of $C^\top \Theta C$ found in the t-th iteration (Liu & Ihler, 2011), $\epsilon_i$ is the positive quantity, $D = \Theta + A$ is the degree matrix of the original graph, and more details are in (Liu & Ihler, 2011).

Using the regularizer $h(\Theta_c)$ for sparsity and scale-free defined in equation 3 and equation 6, the proposed formulation for learning the coarsened graph with structure is

$$\underset{C \in \mathcal{S}_c}{\text{minimize}} \quad f(C) = -\gamma \log \det(C^\top \Theta C + J) + \frac{\lambda}{2} \|C^\top\|_{1,2}^2 + \alpha h(\Theta_C) \tag{8}$$

**Lemma 1.** *Problem (8) is a strictly convex optimization problem.*

*Proof.* The function $-\gamma \log \det(C^\top \Theta C + J) + \frac{\lambda}{2} \|C^\top\|_{1,2}^2$ is a strictly convex function (Kumar et al., 2023), $h(\Theta_c)$ defined in equation 3 and equation 6 are convex functions and the set $\mathcal{S}_C$ is a closed and convex set; thus, problem (8) is a strictly convex optimization problem. □

Since, there does not exist a closed-form solution to the problem (8). To solve it efficiently, we will use the majorization-minimization (MM) framework to obtain easily solvable surrogate functions for objective functions such that the update rule is easily obtained. The surrogate function $g(C|C^{(t)})$ is such that it upper-bounds the objective function $f(C)$ and is tangent to it at the current estimate. By using the first-order Taylor series approximation, a majorized function for $f(C)$ at $C^{(t)}$ can be obtained as (Beck & Pan, 2018; Razaviyayn et al., 2012; Sun et al., 2017):

$$g(C|C^{(t)}) = f(C^{(t)}) + (C - C^{(t)})\nabla f(C^{(t)}) + \frac{L}{2} \|C - C^{(t)}\|^2 \tag{9}$$

where $f(C)$ is $L-$Lipschitz continuous gradient function $L = \max(L_1, L_2, L_3)$ with $L_1, L_2, L_3$ the Lipschitz constants of $-\gamma \log \det(C^\top \Theta C + J)$, $\|C^\top\|_{1,2}^2$, $h(\Theta_c)$ respectively. More details are deferred to the supplementary material. After ignoring the constant term, the majorised problem of (8) is

$$\underset{C \in \mathcal{S}_c}{\text{minimize}} \quad \frac{1}{2} C^\top C - C^\top A. \tag{10}$$

where $A = C^{(t)} - \frac{1}{L} \nabla f(C^{(t)})$. Note that since $C \geq 0$, it implies that $|C_{ij}| = C_{ij}$ hence $\|C^\top\|_{1,2}^2$ is differentiable. Also, the worst case computational complexity is $\mathcal{O}(p^2 k)$, which is due to the matrix multiplication in the gradient of f(C) in equation 8.

**Lemma 2.** *By using KKT optimality condition, the optimal solution of (10) is* $C^{(t+1)} = \left(C^{(t)} - \frac{1}{L} \nabla f\left(C^{(t)}\right)\right)^+$ *where* $(X_{ij})^+ = \max(\frac{X_{ij}}{\|[X^T]_i\|_2}, 0)$ *and* $[X^T]_i$ *is the i-th row of matrix* $X$ *and* $\nabla f(C^{(t)}) = -2\gamma \Theta C^{(t)} (C^{(t)\top} \Theta C^{(t)} + J)^{-1} + \lambda C^{(t)} \mathbf{1} + \nabla h(\Theta_c)$ *where* $\mathbf{1}$ *is all ones matrix of dimension* $k \times k$.

*Proof:* The proof is deferred to the supplementary material. Next, by using the equation 3 and equation 6 as enforcing sparsity and scale-free structure, we obtain sparse coarsened graph(SCG) and scale-free coarsened graph (SFCG), respectively. The gradients $\nabla h(\Theta_c)$ take the following form, for SCG $\nabla h(\Theta_c^{(t)}) = 2\alpha \Theta C^{(t)} C^{(t)\top} \Theta C^{(t)}$ and for SFCG $\nabla h(\Theta_c^{(t)}) = 2AC^{(t)}\delta \mathbf{1}_{k \times k} + 2DC^{(t)}\beta \mathbf{1}_{k \times k}$.

# 4    PROPOSED FRAMEWORK FOR MULTI-COMPONENT & BIPARTITE GRAPH COARSENING VIA LAPLACIAN AND ADJACENCY SPECTRAL CONSTRAINTS

The proposed formulation for learning multi-component coarsened graph via Laplacian spectral constraint(MGC) is

$$\underset{C, \Lambda, U}{\text{minimize}} \quad -\gamma \log \det(\Lambda) + \frac{\lambda}{2} \|C^\top\|_{1,2}^2 + \frac{\beta}{2} \|C^\top \Theta C - U\Lambda U^\top\|_F^2$$
$$\text{subject to} \quad C \in \mathcal{S}_c, \ U^\top U = I, \ \Lambda \in S_\lambda \tag{11}$$

where $C^\top \Theta C$ is the desired Laplacian matrix of coarsened graph which seeks to admit the decomposition $C^\top \Theta C - U\Lambda U^\top$, $\Lambda \in \mathbb{R}^{k \times k}$ is a diagonal matrix containing $\{\lambda_i\}_{i=1}^k$ on its diagonal, and $U \in \mathbb{R}^{k \times k}$ is the matrix satisfying $U^\top U = I$. We are incorporating multi-component structure by enforcing $\{\lambda_i\}_{i=1}^k \in \mathcal{S}_\lambda$. Furthermore, the term $\frac{\beta}{2}\|C^T \Theta C - U\Lambda U^T\|_F^2$ will keep $C^\top \Theta C$ close to $U\Lambda U^\top$ instead of exactly solving the constraint. Note that by choosing sufficiently large $\beta$ can make this relaxation tight. We consider solving for learning an n component graph structure utilizing the constraints in equation 4, where the first n eigenvalues are zero. There are a total of $q = k - n$ non-zero eigenvalues ordered in the given set $\mathcal{S}_\lambda = \{c_1 \leq \lambda_{n+1} \leq \ldots \leq \lambda_k \leq c_2\}$. Collecting the variables as $(C \in \mathbb{R}_+^{p \times k}, \Lambda \in \mathbb{R}^{k \times k}, U \in \mathbb{R}^{k \times k})$, we develop a block MM-based algorithm which updates one variable at a time while keeping the other fixed.

**Update of** $C$: Treating C as variable and fixing $\Lambda$ and $U$,we obtain the following sub-problem for C:

$$\underset{C \in S_c}{\text{minimize}} \quad f(C) = \frac{\lambda}{2}\|C^\top\|_{1,2}^2 + \frac{\beta}{2}\|C^\top \Theta C - U\Lambda U^\top\|_F^2 \tag{12}$$

The function $f(C)$ in problem (12) is a convex function; more details are in the supplementary material. The set $\mathcal{S}_C$ is a closed and convex set; thus (12) is a strongly convex optimization problem. However, there does not exist a closed-form solution to it. To get a closed-form update rule, we will use the MM framework. We have derived the update of C in a similar way as in section 3. The update rule is similar to Lemma 2, where the $\nabla f(C^{(t)}) = \lambda C^{(t)} \mathbf{1} + 2\beta \Theta C^{(t)} (C^{(t)\top} \Theta C^{(t)} - U\Lambda U^\top)$ where $\mathbf{1}$ is all ones matrix of dimension $k \times k$. More details are deferred to the supplementary material.

**Update of** $U$: Treating $U$ as variable and fixing $C$ and $\Lambda$, we obtain the following sub-problem for $U$:

$$\underset{U^\top U = I_q}{\text{minimize}} \quad \frac{\beta}{2}\|C^\top \Theta C - U\Lambda U^\top\|_F^2 = -\frac{\beta}{2}\text{tr}(U^\top C^\top \Theta C U \Lambda) \tag{13}$$

The solution to problem (13) is $U^{t+1} = eigenvectors(C^\top\Theta C)[n+1:k]$ as solved in (Absil et al., 2008; Benidis et al., 2016).

**Update of $\Lambda$:** Treating $\Lambda$ as variable and fixing $C$ and $U$, we obtain the following sub-problem for $\Lambda$:

$$\underset{\Lambda \in S_\lambda}{\text{minimize}} \quad -\gamma \log \det(\Lambda) + \frac{\beta}{2}||C^\top\Theta C - U\Lambda U^\top||_F^2 \tag{14}$$

For ease of notation, we denote the index of non-zero eigen values as $i=1$ to $q$ instead of $i=n+1$ to $k$. We can rewrite problem (14) as:

$$\underset{c_1 \le \lambda_1 \le \lambda_2 \dots \lambda_q \le c_2}{\text{minimize}} \quad -\gamma \sum_{i=1}^q \log \lambda_i + \frac{\beta}{2}||\Lambda - \mathbf{d}||_F^2 \tag{15}$$

where $\lambda = [\lambda_1, \dots \lambda_q]$ and $d = [d_1, \dots, d_q]$ with $d_i$ being the $i$-th diagonal element of $U^\top C^\top \Theta C U$.

**Lemma 3.** *By using KKT condition we can obtain the solution of convex optimization problem (15) is*

$$\lambda_i = \frac{1}{2}(d_i + \sqrt{d_i^2 + 4/\beta}) \qquad \forall i = 1, 2, \dots, q \tag{16}$$

*Proof:* The proof is deferred to the supplementary material.

## 4.1 Bipartite Graph Coarsening via Adjacency Spectral Constraints (BI-GC)

The proposed formulation for learning a Bipartite coarsened graph is

$$\begin{aligned} &\underset{C,\Psi,V}{\text{minimize}} && -\gamma \log \det(C^\top\Theta C + J) + \frac{\lambda}{2}||C^\top||_{1,2}^2 + \frac{\beta}{2}||C^\top AC - V\Psi V^\top||_F^2 \\ &\text{subject to} && C \in \mathcal{S}_c,\ V^\top V = I,\ \Psi \in S_\psi \end{aligned} \tag{17}$$

Suppose there are z number of zero eigenvalues in the set $\mathcal{S}_\psi$. From the symmetry property of the eigenvalues, the zero eigenvalues are positioned in the middle, i.e., in equation 5 the eigenvalues $\psi_{\frac{k-z}{2}+1}$ to $\psi_{\frac{k+z}{2}}$ will be zero. Both $(k+z)$ and $(k-z)$ must be even by the symmetry property. As a consequence, the zero eigenvalues and the corresponding eigenvectors can be dropped from the formulation. Now $\psi \in \mathbb{R}^b$ contains b non-zero eigenvalues and $V \in \mathbb{R}^{k \times b}$ contains the corresponding eigenvectors that satisfy $V^T V = I_b$. The non-zero eigenvalues are required to lie in the set $\mathcal{S}_\psi = \{\psi_i = -\psi_{b+1-i}, c_1 \ge \psi_1 \ge \psi_2 \dots \ge \psi_{b/2} \ge c_2, \forall i = 1, 2, \dots b/2\}$. Where $c_1$ and $c_2 > 0$ are some constants that depend on the graph properties. Collecting the variables as $(C \in \mathbb{R}_+^{p \times k}, \Psi \in \mathbb{R}^{k \times k}, U \in \mathbb{R}^{k \times k})$, we develop a block MM-based algorithm which updates one variable at a time while keeping the other fixed.

**Update of $C$:** Treating C as variable and fixing $\Psi$ and $V$, we obtain the following sub-problem for C:

$$\underset{C \in S_c}{\text{minimize}} \quad -\gamma \log \det(C^\top\Theta C + J) + \frac{\lambda}{2}||C^\top||_{1,2}^2 + \frac{\beta}{2}||C^\top AC - V\Psi V^\top||_F^2 \tag{18}$$

The function $f(C)$ in problem (18) is a convex function, and more details are in the supplementary material. The set $\mathcal{S}_C$ is a closed and convex set; thus (18) is a strongly convex optimization problem. However, there does not exist a closed-form solution to it. To get a closed-form update rule, we will use the MM framework. We have derived the update of C in a similar way as in section 3. The update rule is similar to Lemma 2, where the $\nabla f(C^{(t)}) = -2\gamma\Theta C^{(t)}(C^{(t)\top}\Theta C^{(t)} + J)^{-1} + \lambda C^{(t)}\mathbf{1} + 2\beta AC^{(t)}(C^{(t)\top}AC^{(t)} - V\Psi V^\top)$ where $\mathbf{1}$ is all ones matrix of dimension $k \times k$.

**Update of $V$:** Treating $V$ as variable and fixing $C$ and $\Psi$, we obtain the following sub-problem for $V$:

$$\underset{V^\top V = I_b}{\text{minimize}} \quad \frac{\beta}{2}||C^\top AC - V\Psi V^\top||_F^2 = -\frac{\beta}{2}\text{tr}(V^\top C^\top ACV\Psi) \tag{19}$$

The solution to problem (19) is $V^{t+1} = eigenvectors(C^\top AC)[1 : \frac{k-z}{2}, \frac{k+z}{2} : k]$ as solved in (Absil et al., 2008; Benidis et al., 2016).

**Update of $\Psi$:** The update of $\Psi$ is similar to the update of $\lambda$ without log determinant in problem (14). The detailed proof is in the supplementary material.

---

**Algorithm 1:** Multi-component graph coarsening(MGC) and Bipartitte graph coarsening(BI-GC)

---

**Input:** $\mathcal{G}(\Theta), \beta, \gamma, \lambda$

**1 while** *stopping criteria not met* **do**

**2** | Update $C^{t+1}, U^{t+1}$, and $\Lambda^{t+1}$ for MGC or $C^{t+1}, V^{t+1}$, and $\Psi^{t+1}$ for BI-GC

**3** | $t \leftarrow t + 1$;

**4 end**

**Output:** $C$ and $\Theta_c$

---

## 5 EXPERIMENTS

In this section, we demonstrate the effectiveness of the proposed algorithms by a comprehensive set of experiments conducted on real data sets. We compare the proposed methods for structured graph coarsening against the state-of-the-art methods, GCOND(Jin et al., 2021), SCAL(Huang et al., 2021). However, we considered these state-of-the-art algorithms since they are more recent and outperform existing state-of- the-art graph coarsening approaches. The performance is evaluated through classification accuracy (ACC) and time($\tau$) required to perform coarsening and classification. It has been experimentally verified that proposed methods for structured graph coarsening outperform in classification tasks and time complexity($\tau$).

| Dataset | Nodes | Edges | Features | Classes |
|---|---|---|---|---|
| CORA | 2,708 | 5,429 | 1,433 | 7 |
| CITESEER | 3,327 | 9,104 | 3,703 | 6 |
| DBLP | 17,716 | 52,867 | 1,639 | 4 |
| COAUTHOR CS | 18,333 | 163,788 | 6,805 | 15 |
| PUBMED | 19,717 | 44,338 | 500 | 3 |
| COAUTHOR PHYSICS | 34,493 | 247,962 | 8,415 | 5 |

Table 1: Datasets used in node classification.

| Data set(ACC) | r=k/p | GCOND | SCAL | MGC | BI-GC |
|---|---|---|---|---|---|
| CORA | 0.5 | $81.02 \pm 0.37$ | $82.7 \pm 0.50$ | $\mathbf{87.20 \pm 0.43}$ | $86.26 \pm 0.04$ |
| | 0.3 | $81.56 \pm 0.6$ | $79.42 \pm 1.71$ | $84.56 \pm 1.40$ | $\mathbf{85.15 \pm 0.03}$ |
| CITESEER | 0.5 | $74.28 \pm 1.45$ | $72.0 \pm 0.5$ | $78.80 \pm 1.20$ | $\mathbf{79.69 \pm 0.37}$ |
| | 0.3 | $72.43 \pm 0.94$ | $74.54 \pm 1.34$ | $74.60 \pm 2.31$ | $\mathbf{77.09 \pm 0.24}$ |
| CO-PHY | 0.05 | $93.05 \pm 0.26$ | $73.09 \pm 7.41$ | $\mathbf{94.52 \pm 0.19}$ | $91.63 \pm 0.45$ |
| | 0.03 | $92.81 \pm 0.31$ | $63.65 \pm 9.65$ | $\mathbf{93.64 \pm 0.25}$ | $91.39 \pm 0.35$ |
| PUBMED | 0.05 | $78.16 \pm 0.30$ | $72.82 \pm 2.62$ | $\mathbf{81.89 \pm 0.00}$ | $81.72 \pm 0.48$ |
| | 0.03 | $78.04 \pm 0.47$ | $70.24 \pm 2.63$ | $\mathbf{80.70 \pm 0.00}$ | $80.66 \pm 0.55$ |
| CO-CS | 0.05 | $86.29 \pm 0.63$ | $34.45 \pm 10.07$ | $\mathbf{87.25 \pm 0.90}$ | $84.40 \pm 0.0$ |
| | 0.03 | $86.32 \pm 0.45$ | $26.06 \pm 9.29$ | $\mathbf{86.38 \pm 3.37}$ | $83.41 \pm 0.06$ |
| DBLP | 0.05 | $79.15 \pm 0.20$ | $76.52 \pm 2.88$ | $78.09 \pm 1.88$ | $\mathbf{79.20 \pm 0.07}$ |
| | 0.03 | $78.42 \pm 1.26$ | $75.49 \pm 2.84$ | $74.81 \pm 1.57$ | $\mathbf{78.99 \pm 0.71}$ |

Table 2: The table summarizes the node classification accuracy on real benchmark datasets for the proposed MGC and BI-GC algorithms against the GCOND and SCAL. For small datasets, we have taken coarsening ratio $r = 0.3$ and $0.5$, while for large datasets, we have taken $r = 0.05$ and $0.03$. It is evident that proposed MGC and BI-GC outperform state-of-the-art methods by a significant amount. Furthermore, across a wide range of datasets, the MGC algorithm consistently outperforms or demonstrates comparable performance to the BI-GC algorithm. Additional results with the SCG and the SFCG are provided in the supplementary material.

Furthermore, we have also demonstrated the generalizability of the proposed algorithm by performing node classification on different GNN structures like GCN (Kipf & Welling, 2016), APPNP (Gasteiger et al., 2018), and GAT (Velivcković et al., 2017). Moreover, all the experiments were performed on 16GB RAM GPU(NVIDIA P100) processor. Table 1 shows the statistics of the graph datasets used to perform experiments.

## 5.1 NODE CLASSIFICATION

In this section, we compute the experiments of node classification on real benchmark datasets. For node classification, a GNN model is trained on coarsened graph data, i.e. $\mathcal{G}_c = (\tilde{V}, \tilde{E}, \tilde{X}, \tilde{Y})$. The hyperparameters for graph coarsening and the GNN model are tuned using grid search. The learning and decay rates used in the node classification experiments are 0.01 and 0.0001, respectively. The GNN model is tested on full original graph data $\mathcal{G} = (V, E, X, Y)$ to predict the label of every $p$ node. Then these predicted, and actual labels are used to compute accuracy (ACC). All the results are calculated using 10-fold cross-validation. Also, for the proposed MGC, we have taken components $n$ as the number of classes of the original graph. It is evident in Table 2 and 3 that enforcing structure on the coarsened graph improves the node classification accuracy.

| Data set | GCOND | SCAL | SCG | MGC | GC-BI | Whole Data |
|---|---|---|---|---|---|---|
| CORA | $79.37 \pm 0.4$ | $71.38 \pm 3.6$ | $79.81 \pm 0.3$ | $76.02 \pm 0.9$ | $\mathbf{80.07 \pm 1.20}$ | $89.50 \pm 1.2$ |
| CITESEER | $70.46 \pm 0.4$ | $68.58 \pm 2.3$ | $69.49 \pm 2.1$ | $\mathbf{70.57 \pm 1.2}$ | $68.59 \pm 0.12$ | $78.09 \pm 1.9$ |
| PUBMED | $78.57 \pm 0.2$ | $73.59 \pm 3.5$ | $83.13 \pm 0.1$ | $\mathbf{84.81 \pm 1.5}$ | $81.83 \pm 0.36$ | $88.89 \pm 0.5$ |
| CO-PHY | $92.98 \pm 0.5$ | $86.43 \pm 2.4$ | $81.59 \pm 5.1$ | $\mathbf{94.71 \pm 0.2}$ | $91.84 \pm 0.06$ | $96.22 \pm 0.7$ |
| CO-CS | $87.13 \pm 0.7$ | $55.20 \pm 4.3$ | $90.41 \pm 0.8$ | $\mathbf{91.67 \pm 0.0}$ | $88.08 \pm 0.0$ | $93.32 \pm 0.6$ |
| DBLP | $80.40 \pm 0.9$ | $76.66 \pm 1.7$ | $80.49 \pm 0.1$ | $\mathbf{81.82 \pm 0.6}$ | $80.79 \pm 0.56$ | $85.35 \pm 0.8$ |

Table 3: The table summarizes the node classification accuracy on real datasets for the proposed structured graph coarsening algorithms against the GCOND (Jin et al., 2021), SCAL (Huang et al., 2021) for a coarsening ratio of 0.1. It is evident that the proposed structured graph coarsening algorithm outperforms state-of-the-art methods. However, we have compared only with GCOND and SCAL as these are the most recent technique for graph node classification using the coarsened graph. It is evident that the enforcing structure on the coarsened graph improves the performance significantly.

| Data set(ACC) | r=k/p | SCG | TSM-1 | TSM-2 | TSBI |
|---|---|---|---|---|---|
| CORA | 0.1 | 79.81 | 60.6 | 53.21 | 28.32 |
| | 0.3 | 85.15 | 61.6 | 71.34 | 85.82 |
| CITESEER | 0.1 | 69.29 | 51.93 | 55 | 20.2 |
| | 0.3 | 74.26 | 58 | 63.02 | 75.01 |

Table 4: The table summarizes node classification accuracy for both single-stage and two-stage structured coarsened graph learning methods. Notably, the single-stage structured coarsened graph learning algorithms consistently outperform their two-stage counterparts. In the two-stage multicomponent coarsened graph learning approaches, which is denoted as TSM-1 and TSM-2, the initial stage entails the construction of the graph using the SCG algorithm. Subsequently, in the second stage, the multi-component structure is imposed through the Louvain method(TSM-1)(De Meo et al., 2011) and the MGAE (Multi-Graph Autoencoder)(TSM-2) (Wang et al., 2017) technique. On the other hand, TSBI represents the two-stage bipartite graph learning approach. The coarsened graph is constructed using the SCG algorithm in the first stage. Then, to enforce the bipartite structure, the SGL (Kumar et al., 2020)(Sparse Graph Learning) algorithm is employed.

| Proposed | CORA/(n) | CITESEER/(n) | PUBMED/(n) | CO-PHY/(n) | DBLP/(n) |
|---|---|---|---|---|---|
| MGC | $76.16 \pm 0.1/5$ | $69.82 \pm 1.6/4$ | $82.28 \pm 1.6/1$ | $90.75 \pm 0.3/13$ | $80.77 \pm 0.8/3$ |
| | $76.08 \pm 0.9/7$ | $69.43 \pm 1.3/6$ | $84.81 \pm 1.5/3$ | $91.67 \pm 0.0/15$ | $80.83 \pm 0.9/5$ |
| | $77.62 \pm 2.2/9$ | $69.64 \pm 0.4/8$ | $83.65 \pm 2.4/5$ | $91.08 \pm 0.1/17$ | $81.35 \pm 1.1/7$ |

Table 5: The table summarizes the node classification accuracy on real datasets for the proposed multi-component graph coarsening (MGC) algorithm for different values of component $n$ and coarsening ratio 0.1. It is evident that enforcing multi-component structures on the coarsened graph for different values of component $n$ improves the accuracy.

Moreover, we demonstrated the adaptability of the multicomponent graph coarsening (MGC) algorithm by producing coarsened graphs with different component $n$ values. It is evident that changing the value of n does not decrease classification accuracy as shown in the Table 5. Next, we will illustrate the generalizability of the learning structured coarsened graph from the proposed algorithms

by using different architectures to train the GNN. Specifically, we have used GNN architectures like GCN (Kipf & Welling, 2016), APPNP (Gasteiger et al., 2018), and GAT (Velivcković et al., 2017) to train our GNN and perform the node classification task. Table 6 demonstrates that the proposed methods for learning structured coarsened graph is compatible with different widely used GNN architectures, giving almost similar Node Classification accuracy across all the datasets.

| Data set | GCN | GAT | APPNP |
|---|---|---|---|
| Cora | $79.81 \pm 0.3$ | $76.29 \pm 0.1$ | $79.70 \pm 0.2$ |
| Citeseer | $69.49 \pm 2.1$ | $67.83 \pm 1.3$ | $66.72 \pm 1.2$ |
| Pubmed | $83.13 \pm 0.1$ | $83.53 \pm 0.3$ | $81.90 \pm 0.5$ |
| Co-Phy | $81.59 \pm 5.1$ | $83.43 \pm 1.7$ | $93.88 \pm 0.1$ |
| Co-CS | $87.13 \pm 0.7$ | $88.32 \pm 0.2$ | $90.41 \pm 0.8$ |
| DBLP | $80.09 \pm 0.1$ | $79.01 \pm 1.1$ | $79.05 \pm 0.2$ |

Table 6: Node classification accuracy (%) obtained using different GNN structures like GCN, GAT, and APPNP on different datasets using the proposed SCG algorithm for a coarsening ratio of 0.1. It is evident that the proposed SCG method is suitable for all GNN architecture. However, experiments on different GNN architectures for the remaining proposed algorithms are in the supplementary material.

**Run-time Complexity**: The worst-case computational complexity of SCG, MGC and BI-GC are $\mathcal{O}(p^2 k)$. This section compares the time ($\tau$) required to perform coarsening and node classification by proposed methods against the state-of-the-art algorithm. It is evident in Table 7 that proposed methods for structured graph coarsening are much faster than state-of-the-art methods.

| Dataset($\tau$) | r = k/p | GCOND | SCAL | SCG | MGC | BI-GC | Whole dataset |
|---|---|---|---|---|---|---|---|
| CORA | 0.05 | 329.86 | 27.76 | **2.60** | 2.78 | 3.10 | 2.86 |
| CITESEER | 0.05 | 331.33 | 56.21 | 5.08 | **3.73** | 4.47 | 5.24 |
| PUBMED | 0.05 | 202.04 | 54.09 | **27.99** | 46.09 | 45.68 | 58.85 |
| CO-CS | 0.05 | 1600.32 | 180.16 | **44.45** | 49.80 | 56.23 | 72.31 |

Table 7: This table summarizes the time ($\tau$) in sec. required to perform coarsening and node classification for a coarsening ratio of 0.05. It is evident that the proposed SCG, MGC and Bi-GC are faster than state-of-the-art algorithms. Moreover, the time required to perform coarsening and node classification using the proposed methods is less than that required to perform node classification using the given original graph.

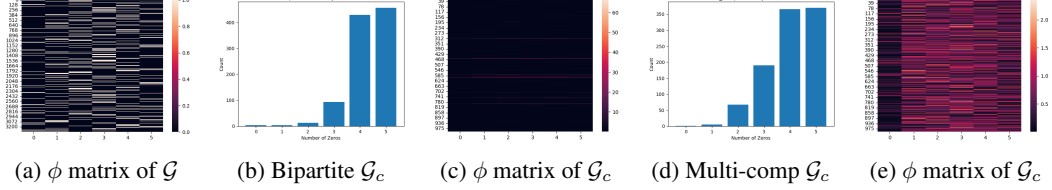

(a) $\phi$ matrix of $\mathcal{G}$    (b) Bipartite $\mathcal{G}_c$    (c) $\phi$ matrix of $\mathcal{G}_c$    (d) Multi-comp $\mathcal{G}_c$    (e) $\phi$ matrix of $\mathcal{G}_c$

Figure 3: The figures presented here depict the $\phi$ matrices of three different graph representations: the original graph, the bipartite coarsened graph, and the multi-component coarsened graph. Additionally, a histogram showcases the sparsity levels within each row of the coarsened graphs, focusing on data from the Citeseer dataset. Notably, these visualizations illustrate a critical point: the sparser the heat map of the coarsened graph, the more informative it becomes. To be specific, the node classification accuracy achieved using the bipartite coarsened graph is $77.11\%$. Meanwhile, the multi-component coarsened graph yields a node classification accuracy of $74.68\%$ for a coarsening ratio of 0.3. The remaining heat maps of $\phi$ matrices are in the supplementary material.

## 6 CONCLUSION

In summary, we have developed an optimization-based framework, structured graph coarsening, that can learn reduced graphs with desirable structures like sparsity, scale-free, bipartite, and multi-component. We have performed a node classification task on the structured coarsened graph, and it is evident that enforcing structure in the coarsened graph increases the accuracy by a significant amount. Moreover, across a wide range of datasets, the MGC algorithm consistently outperforms or demonstrates comparable performance to the BI-GC algorithm. Furthermore, the proposed methods for structured graph coarsening are also suitable for performing tasks on various GNN structures like GCN, APPNP, and GAT. The proposed methods are provably convergent and much faster than the state-of-the-art algorithms.

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
