# OpenReview forum: "Structured Graph Reduction for Efficient GNN"
_ICLR.cc/2024/Conference — Submitted to ICLR 2024_

### Official Review · Reviewer_Nawj · 2023-10-28

**Soundness:** 2 fair
**Presentation:** 2 fair
**Contribution:** 2 fair
**Rating:** 5
**Confidence:** 2

**Summary:**

This paper develops an optimization-based framework for structured graph coarsening. The proposed approach coarsens the graph based on some desirable structural properties such as sparsity, scale-free, bipartite, and multi-component. Experiments have been conducted on different real-world data sets on the node classification task.

**Strengths:**

1) Scalability is an important challenge for GNNs.
2) The optimization procedures seem reasonable.
3) The proposed method is fast.

**Weaknesses:**

Overall, I think the paper solves an important problem, but is not well written enough for publishing.
1) Claiming two baselines from 2021 are the recent state-of-art baselines to be compared with seems not convincing. The paper has mentioned a more recent work in Sec. 2.2, Kumar et al. 2023. It is not clear to me why Kumar et al. 2023 is not compared against in the experiments.
2) Sec. 2 is not well-motivated and not intuitive enough.  For example, why should we use (1)? Why are structural properties such as multi-component and scale-free important in structured graph coarsening?
3) There are several proposed methods based on different desired coarsened graph properties, but it is not clear which is preferred in what scenarios. Also, why not impose all constraints?
4) The experimental results are only between coarsening methods, but it would be better to also compare the performance with the original GNN model without coarsening and show the performance gap.
5) It is not clear what specific GNN is used in Sec. 5.
6) It is not explained why the baselines instead of the proposed methods are boldfaced in the header of Table 7.
7) Typos exist. For example,  the first minus sign after equation (11) should instead be an equal sign I guess?

**Questions:**

1) The paper has mentioned a more recent work in Sec. 2.2, Kumar et al. 2023. Why Kumar et al. 2023 is not compared against in the experiments?
2) In Sec. 2, why should we use (1)? Why are structural properties such as multi-component and scale-free important in structured graph coarsening?
3) Which structural property is preferred in what scenarios? Also, why not impose all constraints?
4) What is the performance gap between the proposed coarsened GNN and the original one?
5) What is the specific GNN used in Sec. 5?
6) Why should the baselines instead of the proposed methods be boldfaced in the header of Table 7?

---

> ### Author Response · Authors · 2023-11-17
> **Response to Reviewer Nawj**
>
> We thank the reviewer for valuable feedback and insightful comments.
>
> **Q1: The paper has mentioned a more recent work in Sec. 2.2, Kumar et al. 2023. Why Kumar et al. 2023 is not compared against in the experiments?**
>
> **A1:** We thank the reviewer for the question. Regarding the paper by **Kumar et al. (2023)** mentioned in Section 2.2, it is essential to note that they did not perform a downstream task such as node classification. Instead, they compared the eigenerror between the original and coarsened graphs. In response to the query, we conducted node classification using the **FGC method proposed by Kumar et al. (2023)** and compared it against our proposed algorithms. The comparison table is illustrated below:
>
> | Data     | r    | FGC             | SCG              | MGC              | BI-GC            | Whole Dataset    |
> |----------|------|------------------|------------------|------------------|------------------|------------------|
> | Cora     | 0.3  | 82.79 $\pm$ 0.23 | **85.65 $\pm$ 1.16** | 84.56 $\pm$ 1.40 | 85.15 $\pm$ 0.03 | 89.5 $\pm$ 1.23  |
> | Citeseer | 0.3  | 71.56$\pm$ 0.45  | 72.67 $\pm$ 1,11 | 74.60 $\pm$ 2.31 | **77.09 $\pm$ 0.24** | 78.09 $\pm$ 1.96 |
> | Pubmed   | 0.05 | 78.46 $\pm$ 0.37 | 78.67 $\pm$ 0.57 | **81.89 $\pm$ 0.01** | 81.72 $\pm$ 0.48 | 88.89 $\pm$ 0.59 |
> | Co-CS    | 0.05 | 86.46 $\pm$ 0.34 | **87.27 $\pm$ 0.95** | 87.25 $\pm$ 0.90  | 84.40 $\pm$ 0.02 | 93.32 $\pm$ 0.62 |
>
> It is evident from the above table that the proposed algorithms outperform all the state-of-the-art graph coarsening algorithms.
>
>
> **Q2: In Sec. 2, why should we use (1)? Why are structural properties such as multi-component and scale-free important in structured graph coarsening?**
>
> **A2:** We thank the reviewer for the question. Equation (1) in section 2 serves as a condition on the mapping matrix (C) to ensure a valid coarsening. This mapping matrix (C) maps nodes in the original graph to the supernodes in the coarsened graph. Each column of C represents a supernode, and every non-zero value in that column corresponds to nodes in the original graph mapped to that specific supernode. Additionally, each row's norm is always equal to 1, signifying that each node is mapped to a single supernode, and columns should be orthogonal such that a node of the original graph should not mapped to two supernodes. The mapping matrix $C$ learned by our algorithms satisfies equation(1).
> To perform the downstream task using the coarsened graph, we need to learn an informative coarsened graph, ensuring that the performance in downstream tasks using the coarsened graph is comparable to that achieved with the original graph. **Ghoroghcgian et al. (2021)** proposed the $\phi$ matrix, defined in Definition 2.1, suggesting that for optimal training of graph neural networks, $\phi$ should be sparse. Our investigation revealed that the $\phi$ matrix calculated from state-of-the-art methods is often not sparse, making it unsuitable for downstream tasks. Empirically, we observed that introducing various structural constraints, such as multicomponent and bipartite structures, and enforcing sparsity in the coarsened graph results in a sparser $\phi$ matrix and, consequently, improved performance. When prior knowledge about the original graph structure is available, imposing the same structure on the coarsened graph enhances downstream task performance. However, in the absence of prior knowledge, empirically, we found that multicomponent structure is a favourable choice for learning a coarsened graph with a sparse $\phi$ matrix, leading to improved performance in node classification tasks.
>
> **Q3: Which structural property is preferred in what scenarios? Also, why not impose all constraints?**
>
> **A3:**  We thank the reviewer for the question. Prior knowledge about a graph's structure can benefit coarsening based on that specific structure. However, without such prior knowledge, the Multi-component structure has proven to be a robust approach, as highlighted in the last paragraph of Section 1, to perform node classification tasks. This observation is further supported by the findings in Table 2, where Multi-component Graph Coarsening (MGC) demonstrates superior or comparable performance compared to other structured coarsening methods.
>  In response to the suggestion of imposing all constraints, we thank the reviewer for the suggestion. Since our framework is optimization-based, all the constraints can be easily imposed, and the overall problem is easily solvable if there is motivation for imposing all the constraints to learn a sparse multicomponent Bipartite coarsened graph. However, it is important to note that imposing all the structures might increase the per-iteration time complexity of the algorithm.

---

> > ### Author Response · Authors · 2023-11-17
> > **Response to Reviewer Nawj**
> >
> > **Q4 What is the performance gap between the proposed coarsened GNN and the original one?**
> >
> > **A4** We thank the reviewer for the question.  The accuracy of node classification for the original graph has been listed in Table 3, section 5, under the column titled **whole dataset**.
> >
> > **Q5: What is the specific GNN used in Sec. 5?**
> >
> > **A5:** We thank the reviewer for the question. We have used GCN architecture for performing the downstream task(Node classification) for all the different structures of the coarsened graph, also mentioned in Section 5.1. Also, we have performed the node classification using the SCG algorithm with GAT and APPNP architecture listed in Table 6. Performance with GAT and APPNP architectures is in the supplementary material for the remaining structures.
> >
> > **Q6: Why should the baselines instead of the proposed methods be boldfaced in the header of Table 7?**
> >
> > **A6:** We thank the reviewer for the question. It was a typing error, and we have updated the paper to reflect the changes as addressed by you.
> >
> > We respectfully ask the reviewers to consider increasing our score if our justification addresses your concerns. We would appreciate further feedback if our changes do not address your concerns.

---

> > > ### Comment · Reviewer_Nawj · 2023-11-20
> > > **Thank you for your response**
> > >
> > > Thank you for your response. I am generally happy with the rebuttal, but I still think the paper requires more work to be ready for publication. I have increased my score from 3 to 5.

---

> > > > ### Author Response · Authors · 2023-11-20
> > > >
> > > > We thank the reviewer for raising our score. We are dedicated to improving the quality of our work and appreciate the constructive feedback received. In view of the reviewer's suggestions for enhancement, we seek your specific feedback and guidance. Thank you for your time and consideration.

---

### Official Review · Reviewer_A6C2 · 2023-10-31

**Soundness:** 2 fair
**Presentation:** 2 fair
**Contribution:** 2 fair
**Rating:** 3
**Confidence:** 4

**Summary:**

This paper studies graph reduction techniques to speedup GNN training. This paper proposes an optimization framework to get reduced graph, which preserves certain desired structural properties. The reduced graph is then used to train GNN parameters. On several benchmark datasets, the proposed method exhibiting improved performance compared to existing coarsening and condensation methods.

**Strengths:**

1. The motivation is cleared explained.
2. The experimental results show the advantages of the proposed method.

**Weaknesses:**

1. The novelty is limited. First, using a graph reduction techniques to speedup GNN training is not new. Second, all the components involved in the coarsening objectives are quite standard, which are widely used for controlling the structures of the solution. Third, the heuristics used to solve the optimization problems are also standard optimization techniques.
2. The datasets used in the experiments are all small: the largest contains 30k nodes. These datasets can be easily trained using full-batch training.
3. For the testing accuracy, there is a notable gap between the proposed methods and whole graph training.
4. In the experiments, the proposed method is compared only against 2 reduction methods. There are numerous coarsening/sparsification/condensation methods, I suggest the authors add more baselines.
5. There is no analyses on the correlation between downstream task performance and the coarsening objectives values, i.e., whether a better coarsening matrix (in terms of the coarsening objectives) results in better classification accuracy.

**Questions:**

1. What data splits are used in the experiments?
2. The authors present the sparsity of $\phi$ in the experiments. I am also interested in the sparsity of the results graph. The original graph is sparse and if the reduced graph is dense, then the training cost could be higher than training on the original (sparse) graph.

---

> ### Author Response · Authors · 2023-11-17
> **Response to Reviewer A6C2**
>
> We thank the reviewer for valuable feedback and insightful comments.
>
> **Q1:  What data splits are used in the experiments?**
>
> **A1:**  We thank the reviewer for this question. We trained the GNN using the coarsened graph, where the label of the coarsened graph is determined by argmax$(C^TY)$, with Y representing the one-hot matrix of labels from the original graph. In the training process, 20% of the labels in Y are masked. Subsequently, testing is conducted on the entire original graph to evaluate the model's performance.
>
>
> **Q2: The authors present the sparsity of $\phi$
>  in the experiments. I am also interested in the sparsity of the results graph. The original graph is sparse and if the reduced graph is dense, then the training cost could be higher than training on the original (sparse) graph.**
>
> **A2:** We thank the reviewer for this question. The coarsened graph learned from the proposed SCG algorithm is sparse  and is demonstrated by the below table:
>
> | Dataset  | Original      | Coarsened   |
> |----------|---------------|-------------|
> | Cora     | (2708, 10556) | (270,1147)  |
> | Citeseer | (3327,9104)   | (327,1016)  |
> | PubMed   | (19717,88648) | (1913,9543) |
>
> In this table, $(x,y)$, where x represents the number of nodes and y represents the number of edges. The table above illustrates a clear correlation: reducing the original graph by a factor of k corresponds to almost same amount of reduction in the number of edges which implies that the sparsity level of the original graph and the coarsened graph is almost the same.
>
> Next, here are the responses to the weaknesses pointed out by the reviewer:
>
> **W1: The novelty is limited. First, using a graph reduction techniques to speedup GNN training is not new. Second, all the components involved in the coarsening objectives are quite standard, which are widely used for controlling the structures of the solution. Third, the heuristics used to solve the optimization problems are also standard optimization techniques.**
>
> **A1:**  We thank the reviewer for the question and for giving us a chance to highlight the novelty of our work. Sandeep Kumar (JMLR, 2020) is an optimization-based framework that learns a graph with various structures by taking the data $X$
>  as input only. **Manoj Kumar (JMLR, 2023)** is also an optimization-based framework that learns a smooth coarsened graph by taking the Laplacian matrix and feature matrix of the original graph as input.  Our algorithm development lies in the intersection of both **Sandeep Kumar (JMLR, 2020) and Manoj Kumar (JMLR, 2023)**. However, the crucial aspect is that different structure plays a different role in performing the downstream task using graph neural network, which has not been investigated yet. Furthermore, this is the first work that enforces specific structures in the coarsened graph and investigates that structure plays an important role while performing downstream tasks using the graph neural network. Moreover, imposing a specific structure into the coarsened graph is a combinatorial hard problem and difficult to solve. We have utilized the spectral constraints to impose the structure defined in Sandeep Kumar (JMLR, 2020) and developed a framework that can enforce four different structures in the coarsened graph. Our framework is simple and adaptable also other than these four structures, structures like Bipartite multicomponent, sparse Bipartite, and tree structure can be easily learned. In this work, we have investigated the role of structure in the coarsened graph for improving the performance of node classification task using the coarsened graph.

---

> ### Author Response · Authors · 2023-11-17
> **Response to Reviewer A6C2**
>
> **W2: The datasets used in the experiments are all small: the largest contains 30k nodes. These datasets can be easily trained using full-batch training.**
>
> **A2:** We thank the reviewer for the question. As per the question by the reviewer, the below table shows the node classification performance on large datasets e.g. **NELL(65755 nodes)** and **OBGN ARXIV(169343 nodes)** for a coarsening ratio of 0.1.
>
> | Dataset    | GCOND           | SCG               | MGC              | BI-GC            |
> |------------|-----------------|-------------------|------------------|------------------|
> | NELL       | OOM             | 85.59 $\pm$ 1.71  | 84.64 $\pm$ 0.08 | **86.28 $\pm$ 0.28** |
> | OGBN-arxiv | 61.3 $\pm$ 0.50 | 55.23 $\pm$ 0.04  | **64.81 $\pm$ 1.91** | 62.87 $\pm$ 3.43 |
>
> Where OOM stands for out of Memory.
> The above table demonstrates that the proposed method outperforms the state-of-the-art graph coarsening method.
>
> **W3: For the testing accuracy, there is a notable gap between the proposed methods and whole graph training.**
>
> **A3:** We thank the reviewer for the question. We agree with the reviewer that there is a gap between the proposed methods and whole graph training. However, this gap is significantly less as compared to other state-of-the-art methods.
>
> **W4: In the experiments, the proposed method is compared only against 2 reduction methods. There are numerous coarsening/sparsification/condensation methods, I suggest the authors add more baselines.**
>
> **A4:** We thank the reviewer for the question. we have also compared our proposed algorithms with the most recent graph coarsening method **FGC(Manoj Kumar (JMLR, 2023))** as illustrated in below table:
>
> | Data     | r    | FGC              | SCG              | MGC              | BI-GC            | Whole Dataset    |
> |----------|------|------------------|------------------|------------------|------------------|------------------|
> | Cora     | 0.3  | 82.79 $\pm$ 0.23 | **85.65 $\pm$ 1.16** | 84.56 $\pm$ 1.40 | 85.15 $\pm$ 0.03 | 89.5 $\pm$ 1.23  |
> | Citeseer | 0.3  | 71.56$\pm$ 0.45  | 72.67 $\pm$ 1,11 | 74.60 $\pm$ 2.31 | **77.09 $\pm$ 0.24** | 78.09 $\pm$ 1.96 |
> | Pubmed   | 0.05 | 78.46 $\pm$ 0.37 | 78.67 $\pm$ 0.57 | **81.89 $\pm$ 0.01** | 81.72 $\pm$ 0.48 | 88.89 $\pm$ 0.59 |
> | Co-CS    | 0.05 | 86.46 $\pm$ 0.34 | **87.27 $\pm$ 0.95** | 87.25 $\pm$ 0.90  | 84.40 $\pm$ 0.02 | 93.32 $\pm$ 0.62 |
>
> It is evident in the above table that our proposed algorithms outperform the state-of-the-art methods.
>
> **W5: There is no analyses on the correlation between downstream task performance and the coarsening objectives values, i.e., whether a better coarsening matrix (in terms of the coarsening objectives) results in better classification accuracy.**
>
> **A5** We thank the reviewer for this question. We have analysed that the downstream task performance depends on the sparsity of $\phi$ matrix. The more sparser the $\phi$ matrix, the more the node classification accuracy will be, which is supported by Figure 3 in the paper.  More detail about $\phi$ matrix is in section 2 of the manuscript.
>
> We respectfully ask the reviewers to consider increasing our score if our justification addresses your concerns. We would appreciate further feedback if our changes do not address your concerns.

---

> > ### Author Response · Authors · 2023-11-21
> >
> > Dear Reviewer,
> >
> > We would like to thank the reviewer once again for dedicating your time to review our paper and for providing us with encouraging feedback. Could you please inform us if your concerns have been adequately addressed? Your insightful comments are highly valued, and we are fully dedicated to enhancing the quality and comprehensiveness of our work.
> >
> > Best Regards, The authors

---

> > > ### Comment · Reviewer_A6C2 · 2023-11-22
> > > **Response to author rebuttal**
> > >
> > > Thanks for the response. After reading the response and comments from other reviewers, I still think the novelty is limited and the performance is not convincing. I decide to keep my score.

---

### Official Review · Reviewer_EfFB · 2023-11-01

**Soundness:** 2 fair
**Presentation:** 3 good
**Contribution:** 2 fair
**Rating:** 5
**Confidence:** 4

**Summary:**

The paper presents a unified optimization-based framework for graph coarsening, addressing the computational challenges of processing large-scale graph data. Previous methods cannot coarsen graphs while preserving desired properties like sparsity, scale-free, bipartite, and multi-component structures. This submission introduces a unified solution leveraging block majorization-minimization and spectral regularization to enforce these structures during the coarsening process efficiently. The proposed method is validated through experiments on some graph benchmark datasets, demonstrating its effectiveness in preserving structural integrity in coarsened graphs. This approach also reveals a robustness that can simplify graph analysis and uncover meaningful insights, highlighting its potential for real-world applications.

**Strengths:**

1. The paper introduces a unified optimization-based framework for graph coarsening that efficiently preserves desirable properties such as sparsity, scale-free, bipartite, and multi-component structures. This approach is based on a previous study but applies to a broader setting.

2. The framework overcomes the limitations of existing methods, which often fail to maintain these properties during coarsening, potentially enhancing the performance of downstream graph-based machine learning tasks.

3. The framework's effectiveness is empirically validated through extensive experiments on real-world benchmark datasets. It demonstrates improved accuracy and efficiency in node classification tasks across various Graph Neural Network (GNN) architectures, setting a new standard for graph coarsening techniques.

**Weaknesses:**

There are several crucial issues:

1. The authors have delineated their contributions within the domain of graph coarsening; however, there's a need for greater specificity and clarity, particularly concerning the distinction from prior works like those of Manoj Kumar (JMLR, 2023) and Sandeep Kumar (JMLR, 2020). The authors need to explain what unique aspects of their framework set it apart from these previous studies. Further elaboration is required in Section 2 to clarify how this work advances the field beyond the current state-of-the-art. This clarification will not only strengthen the paper but also assist readers in understanding the novel contributions made by the authors.

2. The time complexity of the proposed framework is quadratic to the size of the graph $O(n^2k)$ where $n$ is the number of nodes. It will be impractical if one wants to apply this technique to large-scale graphs.

3. The experiments were tested on all small graphs. It would be more interesting if very large graphs were used, such as graphs in OGB datasets.

**Questions:**

Some key questions are:
Here are the refined questions:

1. The reviewer notes that the submission lacks a clear exposition of its novel contributions, especially when contrasted with prior works such as those by Manoj Kumar (JMLR, 2023) and Sandeep Kumar (JMLR, 2020). Could the authors specifically articulate the unique and novel aspects of their approach that advance the state-of-the-art in graph coarsening?

2. While applying convex optimization methods to graph coarsening is theoretically appealing, concerns arise regarding scalability, as time complexity may increase quadratically with the number of nodes. This appears to conflict with the primary objective of graph coarsening, which is to reduce computational load. Can the authors address the apparent contradiction and clarify how their algorithms maintain or improve upon the time complexity compared to existing graph coarsening methods?

3. The datasets employed in validating the proposed technique are relatively small-scale. For a comprehensive evaluation of the proposed method's efficacy, would the authors consider applying their approach to larger-scale graphs where the benefits of coarsening could be more pronounced and the technique's scalability and effectiveness more rigorously assessed?

---

> ### Author Response · Authors · 2023-11-17
> **Response to Reviewer EfFB**
>
> We thank the reviewer for valuable feedback and insightful comments.
>
> **Q1: The reviewer notes that the submission lacks a clear exposition of its novel contributions, especially when contrasted with prior works such as those by Manoj Kumar (JMLR, 2023) and Sandeep Kumar (JMLR, 2020). Could the authors specifically articulate the unique and novel aspects of their approach that advance the state-of-the-art in graph coarsening?**
>
> **A1:** We thank the reviewer for the question and for giving us a chance to highlight the novelty of our work. **Sandeep Kumar (JMLR, 2020)** is an optimization-based framework that learns a graph with various structures by taking the data $X$ as input only. **Manoj Kumar (JMLR, 2023)** is also an optimization-based framework that learns a smooth coarsened graph by taking the Laplacian matrix and feature matrix of the original graph as input. We agree that based on the methodology, our algorithm development lies in the intersection of **both Sandeep Kumar (JMLR, 2020) and Manoj Kumar (JMLR, 2023)**. However, the crucial aspect is that different structure plays a different role in performing the downstream task using graph neural network, which has not been investigated yet. Furthermore, this is the first work that enforces specific structures in the coarsened graph and investigates that structure plays an important role while performing downstream tasks using the graph neural network. Moreover, imposing a specific structure into the coarsened graph is a combinatorial hard problem and difficult to solve. We have utilized the spectral constraints to impose the structure defined in Sandeep Kumar (JMLR, 2020) and developed a framework that can enforce four different structures in the coarsened graph. Our framework is simple and adaptable also other than these four structures, structures like Bipartite multicomponent, sparse Bipartite, and tree structure can be easily learned. In this work, we have investigated the role of structure in the coarsened graph for improving the performance of node classification task using coarsened graph.
>
> **Q2: While applying convex optimization methods to graph coarsening is theoretically appealing, concerns arise regarding scalability, as time complexity may increase quadratically with the number of nodes. This appears to conflict with the primary objective of graph coarsening, which is to reduce computational load. Can the authors address the apparent contradiction and clarify how their algorithms maintain or improve upon the time complexity compared to existing graph coarsening methods?**
>
> A2: We thank the reviewer for the question regarding the per iteration complexity of our algorithm, which is $\mathcal{O}(p^2k)$ due to matrix multiplications. It is crucial to note that the mapping matrix $C$ exhibits high sparsity, containing only $p$ non-zero elements. In the broader context, graphs often demonstrate sparsity, resulting in a sparser Laplacian matrix. To enhance computational efficiency, by leveraging sparse matrix multiplication instead of the conventional matrix multiplication, the per iteration complexity can be significantly reduced to $\mathcal{O}(ab)$, where $'a'$ represents the number of non-zero entries in matrix $A$, and $'b'$ signifies the number of non-zero entries in matrix $B$. For example, **SciPy sparse** module in Python can be directly used for sparse matrix multiplication.
>
>
>
>
> Next, we will compare the time complexity of our proposed algorithms and state-of-the-art techniques as suggested by the reviewer. For instance, the deep learning-based method **GCOND** exhibits a time complexity of $\mathcal{O}(Lk^2d + Lkd)$, where $L$ is the number of GCN layers, and $d$ represents the number of hidden units. In contrast, the most recent optimization-based graph coarsening technique, **FGC (Manoj Kumar, JMLR, 2023)**, has a per-iteration time complexity of $\mathcal{O}(7p^2k+5k^2p+5kpn+k^3)$. Similarly, SCG has a per-iteration time complexity of $\mathcal{O}(3p^2k+2k^2p+3kpn+k^3)$, while **BI-GC and MGC** have $\mathcal{O}(4p^2k+2k^2p+3kpn+k^3)$ which demonstrate that our proposed algorithms are faster than state of the art graph coarsening techniques.
>
>
> Furthermore, consider the input graph with $p$ nodes, $E_1$ edges and a feature vector of size $n$ for each node. If the GCN has $L$ layers, the total time complexity to perform node classification is $\mathcal{O}(Lp^2n+LpE_1n)$. However, the time complexity to perform coarsening as well as node classification is $\mathcal{O}(p^2k+Lk^2n+LkE_2n)$ where k is the number of nodes of the coarsened graph and $E_2$ is the number of edges of the coarsened graph. Since $(p>>k)$ and $(E_1>>E_2)$ and if $(k<n)$, then the time complexity to perform coarsening and node classification is less as compared to performing classification using the original graph.

---

> ### Author Response · Authors · 2023-11-17
> **Response to Reviewer EfFB**
>
> **Q3: The datasets employed in validating the proposed technique are relatively small-scale. For a comprehensive evaluation of the proposed method's efficacy, would the authors consider applying their approach to larger-scale graphs where the benefits of coarsening could be more pronounced and the technique's scalability and effectiveness more rigorously assessed?**
>
> **A3** We thank the reviewer for the question. As per the suggestion by the reviewer, the below table shows the node classification performance on large datasets e.g. **NELL(65755 nodes)** and **OBGN ARXIV(169343 nodes)** for a coarsening ratio of 0.1.
>
> | Dataset    | GCOND           | SCG               | MGC              | BI-GC            |
> |------------|-----------------|-------------------|------------------|------------------|
> | NELL       | OOM             | 85.59 $\pm$ 1.71  | 84.64 $\pm$ 0.08 | **86.28 $\pm$ 0.28** |
> | OGBN-arxiv | 61.3 $\pm$ 0.50 | 55.23 $\pm$ 0.04  | **64.81 $\pm$ 1.91** | 62.87 $\pm$ 3.43 |
>
> Where OOM stands for out of Memory. The above table demonstrates that the proposed method outperforms the state-of-the-art graph coarsening method.
>
> We respectfully ask the reviewers to consider increasing our score if our justification addresses your concerns. We would appreciate further feedback if our changes do not address your concerns.

---

> > ### Author Response · Authors · 2023-11-21
> >
> > Dear Reviewer,
> >
> > We would like to thank the reviewer once again for dedicating your time to review our paper and for providing us with encouraging feedback. Could you please inform us if your concerns have been adequately addressed? Your insightful comments are highly valued, and we are fully dedicated to enhancing the quality and comprehensiveness of our work.
> >
> > Best Regards, The authors

---

### Official Review · Reviewer_nDtt · 2023-11-05

**Soundness:** 3 good
**Presentation:** 2 fair
**Contribution:** 3 good
**Rating:** 6
**Confidence:** 3

**Summary:**

The paper focuses on graph coarsening to reduce a large graph to a smaller tractable graph effectively. A unified optimization framework is introduced in the paper for learning coarsened graphs with desirable structures and properties. The approaches to efficiently solving the proposed framework are further presented in the paper. Extensive experiments are also conducted on various real benchmark datasets to validate the effectiveness of the proposed framework.

**Strengths:**

S1. Graph coarsening is an important problem in reducing a large graph to a smaller tractable graph effectively.

S2. The paper proposes a unified optimization framework for learning coarsened graphs with desirable structures and properties, including sparse graphs, scale-free graphs, multi-component graphs, and bipartite graphs.

S3. Experimental results show that the proposed method outperforms the state-of-the-art graph coarsening methods in terms of node classification accuracy.

**Weaknesses:**

W1. The benchmark datasets are all relatively small. Considering that the primary aim of graph coarsening techniques is to downsize large-scale graphs, validating the model's performance on larger datasets would lend more credibility to the results.

W2. Some experimental results require further clarification. For instance:

- In Table 7, the time required for coarsening and node classification using the proposed methods is not significantly less than that needed for node classification using the original graph. In the worst case (e.g., the BI-GC model on the CORA dataset), the time cost for the former even exceeds the latter. However, graph coarsening techniques are designed to reduce large-scale original graphs and enhance the scalability of existing GNN models. These empirical results seem to contradict the core motivations behind graph coarsening, necessitating further clarification.

- In Tables 1, 2, and 3 provided in the supplementary material, it would be beneficial to include node classification results using the original graph for comparison.

W3. The paper's presentation needs improvements. The literature review part is too short and not very informative. Moreover, the manuscript contains some sentences that are unclear and could lead to ambiguity. For instance:

- The last paragraph of Section 1 is somewhat vague and needs further clarification. While the authors emphasize the significance of adopting the multiple-component method in graph coarsening, the main contribution of this paper is the introduction of a generalized optimization framework that supports four kinds of graph structures simultaneously. The reasons for focusing on the other three structures – sparse graphs, scale-free graphs, and bipartite graphs – merit further explanation.

- Page 2: $C^T$ -> $C^\top$

- Page 3: Where $gdet(\Theta_c)$ denotes -> where $gdet(\Theta_c)$ denotes

- Table 3: Wh.da. -> Whole dataset

**Questions:**

Q1. It is noteworthy that most scale-free graphs also fall into the category of sparse graphs. Could the authors elaborate on the rationale behind discussing these two types of graphs separately?

---

> ### Author Response · Authors · 2023-11-17
> **Response to Reviewer nDtt**
>
> We thank the reviewer for valuable feedback and insightful comments.
>
> **W1: The benchmark datasets are all relatively small. Considering that the primary aim of graph coarsening techniques is to downsize large-scale graphs, validating the model's performance on larger datasets would lend more credibility to the results.**
>
> **A1**:  We thank the reviewer for the question.  As per the suggestion by the reviewer, below table shows the node classification performance on large datasets e.g. **NELL(65755 nodes)** and **OBGN ARXIV(169343 nodes)** for a coarsening ratio of 0.1.
>
> | Dataset    | GCOND           | SCG               | MGC              | BI-GC            |
> |------------|-----------------|-------------------|------------------|------------------|
> | NELL       | OOM             | 85.59 $\pm$ 1.71  | 84.64 $\pm$ 0.08 | **86.28 $\pm$ 0.28** |
> | OGBN-arxiv | 61.3 $\pm$ 0.50 | 55.23 $\pm$ 0.04  | **64.81 $\pm$ 1.91** | 62.87 $\pm$ 3.43 |
>
> Where OOM stands for out of Memory. The above table demonstrates that the proposed method outperforms the state-of-the-art graph coarsening method.
>
> **W2: Some experimental results require further clarification. For instance:**
>
> **<i> In Table 7, the time required for coarsening and node classification using the proposed methods is not significantly less than that needed for node classification using the original graph. In the worst case (e.g., the BI-GC model on the CORA dataset), the time cost for the former even exceeds the latter. However, graph coarsening techniques are designed to reduce large-scale original graphs and enhance the scalability of existing GNN models. These empirical results seem to contradict the core motivations behind graph coarsening, necessitating further clarification.**
>
> **<ii> In Tables 1, 2, and 3 provided in the supplementary material, it would be beneficial to include node classification results using the original graph for comparison.**
>
> **A2 <i>**: We thank the reviewer for the question. Consider the input graph with $p$ nodes, $E_1$ edges and a feature vector of size $n$ for each node. If the GCN has $L$ layers, the total time complexity for node classification is $\mathcal{O}(Lp^2n+LpE_1n)$. However, the time complexity to perform coarsening as well as node classification is $\mathcal{O}(p^2k+Lk^2n+LkE_2n)$ where $k$ is the number of nodes of the coarsened graph and $E_2$ is the number of edges of the coarsened graph. Since $(p>>k)$ and $(E_1>>E_2)$ and if $(k<n)$, then the time complexity to perform coarsening and node classification is less as compared to performing classification using the original graph.
>
> For smaller datasets, the time needed for coarsening and subsequent node classification using our proposed methods may not exhibit a substantial reduction compared to node classification on the original graph. However, as the dataset size scales up, a noteworthy reduction in time becomes apparent, as exemplified by the CO-CS dataset. Moreover, it is crucial to note that the benefits extend beyond time efficiency. The space complexity is also considerably reduced during the GNN training on the coarsened graph.
>
> **A2 <ii>:** We thank the reviewer for the suggestion. We have updated the supplementary as per the suggestion.
>
> **W3. The paper's presentation needs improvements.**
>
> **A3:** We thank the reviewer for the suggestion. We have updated the manuscript as per the reviewer's suggestion.
>
>
> Next, here is the answer to the question asked by the reviewer:
>
>
>  **Q1. It is noteworthy that most scale-free graphs also fall into the category of sparse graphs. Could the authors elaborate on the rationale behind discussing these two types of graphs separately?**
>
> **A1**: We thank the reviewer for the question. We agree with the reviewer that most of the scale-free graphs fall into the category of sparse graphs. However, sparse graph is the broader class of graph. It is important to note that sparsity alone does not guarantee the scale-free characteristic. Additional regularization, as defined in equations (6) and (7), is necessary to ensure the scale-free property.
>
>
> We respectfully ask the reviewers to consider increasing our score if our justification addresses your concerns. We would appreciate further feedback if our changes do not address your concerns.

---

> > ### Author Response · Authors · 2023-11-21
> >
> > Dear Reviewer,
> >
> > We would like to thank the reviewer once again for dedicating your time to review our paper and for providing us with encouraging feedback. Could you please inform us if your concerns have been adequately addressed? Your insightful comments are highly valued, and we are fully dedicated to enhancing the quality and comprehensiveness of our work.
> >
> > Best Regards, The authors

---

### Meta-Review · Area_Chair_Z7oT · 2023-12-05

**Metareview:**

The paper introduces an optimization-based framework for graph coarsening, aiming to efficiently reduce large graphs while preserving key structural properties like sparsity, scale-free, bipartite, and multi-component structures. Utilizing techniques such as block majorization-minimization and spectral regularization, the framework ensures the integrity of these structures during the coarsening process. Extensive experiments on various benchmark datasets demonstrate the method's effectiveness in maintaining structural integrity in coarsened graphs, enhancing graph neural network training, and offering potential for practical applications in graph analysis.

While the proposed method shows promising results, the reviewers have identified several weaknesses that need to be addressed:

1. The paper's novelty is limited, as it employs standard graph reduction techniques and optimization methods for accelerating GNN training, and it lacks distinct differentiation from prior works, such as those by Manoj Kumar (JMLR, 2023) and Sandeep Kumar (JMLR, 2020); this necessitates further specificity and clarification regarding its unique contributions to the field of graph coarsening.
2. Some experimental results in the paper require further clarification.
3. The presentation of the paper needs improvement.

Based on these weaknesses, we recommend rejecting this paper. We hope this feedback helps the authors improve their paper.

**Justification For Why Not Higher Score:**

Three reviewers believed the paper should be rejected, while the other did not champion the paper during the discussion phase.

**Justification For Why Not Lower Score:**

N/A

---

### Decision · Program_Chairs · 2024-01-16

Reject